# The limitations of phenotype prediction in metabolism

**Pablo Yubero**, **Alvar A. Lavin**, **Juan F. Poyatos** *

Logic of Genomic Systems Lab, CNB-CSIC, Madrid, Spain

* jpoyatos@cnb.csic.es

## Abstract

Phenotype prediction is at the center of many questions in biology. Prediction is often achieved by determining statistical associations between genetic and phenotypic variation, ignoring the exact processes that cause the phenotype. Here, we present a framework based on genome-scale metabolic reconstructions to reveal the mechanisms behind the associations. We calculated a polygenic score (PGS) that identifies a set of enzymes as predictors of growth, the phenotype. This set arises from the synergy of the functional mode of metabolism in a particular setting and its evolutionary history, and is suitable to infer the phenotype across a variety of conditions. We also find that there is optimal genetic variation for predictability and demonstrate how the linear PGS can still explain phenotypes generated by the underlying nonlinear biochemistry. Therefore, the explicit model interprets the black box statistical associations of the genotype-to-phenotype map and helps to discover what limits the prediction in metabolism.

## Author summary

Predicting phenotypes from genotypes is fundamental in biology. The prevalent method, quantitative genetics, leverages genetic and phenotypic variation data to create predictive statistical tools like polygenic scores–weighted sums of phenotype-linked alleles. However, this approach overlooks the nonlinear mechanisms that significantly contribute to the genesis of the phenotype. In this work, we used comprehensive genome-wide metabolic models as a direct genotype-to-phenotype map that enables us to scrutinize the mechanistic basis behind statistically forecasting growth (biomass). We introduce a procedure within this framework for generating the necessary genetic and phenotypic diversity essential for calculating polygenic scores and assess the impact of the extent of genetic variation and environmental factors that curtail predictability. Our discoveries underscore that the structure of the polygenic score relies on the interplay among three pivotal components: the functional mode governing phenotype generation, the imprint of evolutionary history on essential parameters, and the genetic architecture at play. Our study exemplifies how integrating prediction and interpretation can yield a broader understanding of how genetic information and phenotypic outcomes interact. This approach transcends linear assumptions,

**Data Availability Statement:** Data and code for this work is available at Zenodo https://zenodo.org/record/6550708). The main code used to generate quantitative mutations is available at GitHub (https://github.com/pyubero/quantitative_mutations).

**Funding:** This work was supported by grant PID2019-106116RB-I00 (P.Y. and J.F.P.), partially by Ph.D. fellowship BES-2016-079127 (P.Y.), and the program Severo Ochoa Center of Excellence (A. A.L.) from the Spanish Ministerio de Ciencia e Innovación and the European Social Fund. The funders had no role in study design, data collection and analysis, decision to publish, or preparation of the manuscript.

**Competing interests:** The authors have declared that no competing interests exist.

enriching our insight into the complex nonlinear nature of genotype-phenotype relationships.

## Introduction

The variational method of quantitative genetics represents a traditional approach to understanding the heritable biological factors that specify the phenotype. Its objective is to establish statistical associations between the genetic and phenotypic variation observed within a given population [1]. When this *genotype-to-phenotype* (GP) map is determined, then it is possible to develop tools that infer the phenotype of individuals based solely on their genetic sequence [2].

Valid as they are, these statistical associations are highly dependent on trait characteristics, population context, and environmental conditions in which they are identified [3]. Thus, they represent a kind of *black box* expectation that does not provide information about the processes that cause a particular phenotype [4]. This absence of mechanism has both basic and applied implications.

From the basic point of view, many characteristics that define the genetic architecture of the phenotypes (dominance, epistasis, etc.), although they have a clear variational definition, present a less clear mechanistic interpretation [5, 6]. An interpretation that should also help explain how the functional non-linearity that seems dominant in many biological systems limits the power of –linear– statistical procedures [7].

From an applied point of view, consider, for example, the case of genome-wide association studies (GWAS) in humans. The original purpose of GWAS was to identify the causal genetic determinants of complex phenotypes, including diseases. This plan turned out to be more complicated than expected [8], with recent studies emphasizing the complex pleiotropic regulation of most human traits [9, 10]. Similarly, although specific predictive tools are available, for example, the development of polygenic scores to indicate a predisposition to disease [11], we still do not understand the biological rationale behind their successes and failures.

Indeed, uncovering the mechanistic insights behind these GP associations has proven to be a significant challenge, due in part to the large number of accepted causal elements that are distinguished for most phenotypes. For example, not long ago, human quantitative traits were linked to only a few strong-effect genes; a hypothesis that is now abandoned [9, 10, 12]. A second factor is that natural selection weakens the impact of the *a priori* stronger determinants [13]. Most significant of all is the absence of an underlying physiological or developmental model to explain the appearance of the phenotype [4, 14]. Therefore, it is interesting to examine situations where an explicit model replaces the black box so that we can clarify the determinants behind the associations.

There have been several attempts in this regard. Plant biology has pioneered work to connect gene network modeling with quantitative genetics, for example, in the prediction of flowering time [15]. Other computational efforts to relate explicit phenotypic patterns and genetic variation include cases of foliate-mediated one-carbon metabolism [16], single heart cells [17], or tooth development [18].

Here, our primary goal is to investigate the mechanistic interpretation of the associations derived from a polygenic score. To that end, we consider a genome-scale metabolic model as an explicit GP map. These models contain all known metabolic reactions in an organism and the genes encoding each enzyme and allow the prediction of metabolic phenotypes in situations where genetics and the environment can be controlled (Methods). We will specifically consider growth (biomass) as the phenotype of interest.

To analyze the emergence of the statistical associations in the context of metabolism, we present a procedure to generate the genetic and phenotypic variation required to calculate a polygenic score. This polygenic score is meant to demonstrate the challenges to understand the GP map, rather than to optimize predictions in true populations. Our analysis clarifies why individual genes act as predictors in this score and how *intrinsic* characteristics, that is, of the GP metabolic map, such as pleiotropy or epistasis, and *extrinsic*, that is, aspects of the population and the environment, are combined to determine the strength of predictability. To do so, we highlight the essential role of the relationship between the functional mode and baseline activity of wild-type metabolism (which we term evolutionary history). More generally, we identify principles that lead to better prediction.

## Results

Quantitative genetics employs the genetic variation that exists in natural populations to estimate statistical associations with phenotypes using a reference, or training, population of known phenotype. Associations are quantified by so-called polygenic scores (PGS, Fig 1A), which are ultimately used to predict the effect of many specific genetic variants on an individual's phenotype.

Our first goal is to emulate this scenario using an in-silico metabolism of *Saccharomyces cerevisiae*. The novelty here is that we can *design* genetic variation by generating a population of metabolisms in which each member exhibits different alleles determining contrasting gene (expression) dosages. Subsequently, the dosages are quantitatively interpreted in the model by

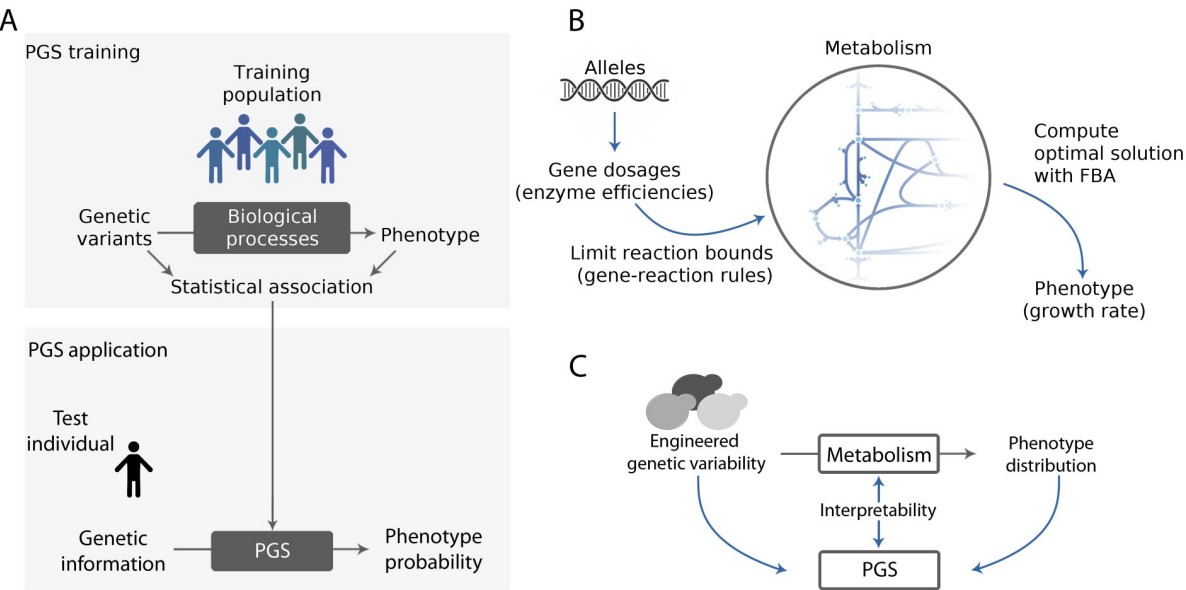

**Fig 1. Metabolic reconstructions provide an explicit GP map that allows us to interpret the black box of phenotype prediction.** A: The statistical association of genetic and phenotypic variants in a training population defines a *polygenic score* (PGS) operating as a black box that bypasses all underlying biological processes. Once this is obtained, the PGS can provide the probability that a test individual exhibits a specific trait value, given its genetic information. B: We benefited from the metabolic reconstruction of *S. cerevisiae* to generate an in-silico population of yeast metabolisms. Genetic variation in this population is modeled by the effect of different alleles on the efficiency of each enzyme –gene expression dosage– which, in turn, limits the maximum flux through its associated reactions according to the gene-reaction rules (GRR; S1 Fig describes a specific example). Given these constraints, flux balance analysis (FBA) calculates the growth rate (phenotype) of each individual in the population. In general, this protocol provides both genetic and phenotypic distributions. C: We quantify the statistical associations between these distributions to determine a PGS for the growth rate. The availability of an underlying (metabolic) model allows us to open the black box of phenotype prediction to investigate its limitations.

Gene Reaction Rules (GRR): Boolean relationships between enzymes that define which and how they participate in the reactions (Fig 1B, Methods and S1 Fig). These rules cover the existence of isozymes and coenzymes and thus contribute towards a more realistic metabolic model.

The resulting calculation represents a decrease in enzyme efficiency from a maximum value in line with previous work [5, 19–21]. To calculate this maximum value, we simulated growth under a number of environmental and genetic conditions that represents a possible *evolutionary history* of yeast metabolism (Methods and S1 Fig). The maximum flux exhibited by each reaction across all of those conditions designates a reference flux for the genetic variation.

The variation created in this way leads to individual differences in any potential metabolic trait, but we focus on the rate of biomass production corresponding to the growth rate that is computed through flux balance analysis (FBA, Fig 1B; see [22] and S1 Text, for a concise introduction to FBA). Thus, the whole procedure produces a dataset of both genetic and phenotypic variations in the context of a metabolic model, which we can dissect to explain how the system works as a whole (Fig 1C).

## A small subset of genes predicts growth within a metabolic polygenic score

We use the described data set to derive a multidimensional PGS. The different dosages induce variations in growth and in many metabolic fluxes when the metabolism function is calculated using FBA. By fixing the growing medium in standard nutrients, we controlled for environmental effects on the phenotype (Fig 2A and 2B and S2 Fig). Also, our approach ensures coherent solutions within a population as all flux distributions are similar among all individuals growing in the same medium (S2 Fig). Then, we obtain a PGS intended to predict the individual growth rate. Fig 2C compares the predicted growth rate with those values obtained with FBA for this training data set. The PGS infers the phenotype with a $R^2_{\mathrm{train}} \sim 0.27$, hereafter abbreviated $R^2$, and also limits possible statistical overfitting by using an arbitrarily large training population and cross validation in the training process (Methods). The arguably limited value of $R^2$ is in line with the predictive ability in other phenotypic systems. In the following, we focus on understanding, rather than maximizing prediction in this particular case.

The derived PGS includes all metabolic genes and their corresponding effect sizes, $\beta$ (Methods). We identified 85 non-null effect genes, 32 of which are comparatively large ($|\beta| > 0.01$; Fig 2D and 2E). The latter impacts 61 metabolic reactions (out of 1148). That most of the fluxes, in the standard medium considered, are inactivated explains this large number of genes with no effect (individual metabolisms within the population typically show $\sim 73\%$ of unused fluxes). In addition, the set of active reactions characterizes a distinct metabolic *functional mode* (a relevant notion in what follows). From now on, we will focus on the subset of genes with larger effect sizes.

## Few metabolic functions limit growth

What kind of functions do predictor genes implement? One might think that the predictors are distributed among all metabolic activities in the sense of universal pleiotropy [19]. However, we only found a few predictor-enriched metabolic subsystems (Methods, S3 Fig and Table A in S1 Text). These subsystems specifically involve the production of biomass precursors. This group of metabolites feeds the biomass reaction, which defines the architecture of the trait –in this case growth– in metabolic reconstructions (S4 Fig shows its stoichiometry; this incorporates, for example, the crucial role of amino acids and phospholipids for protein synthesis and the cell membrane, respectively, etc.).

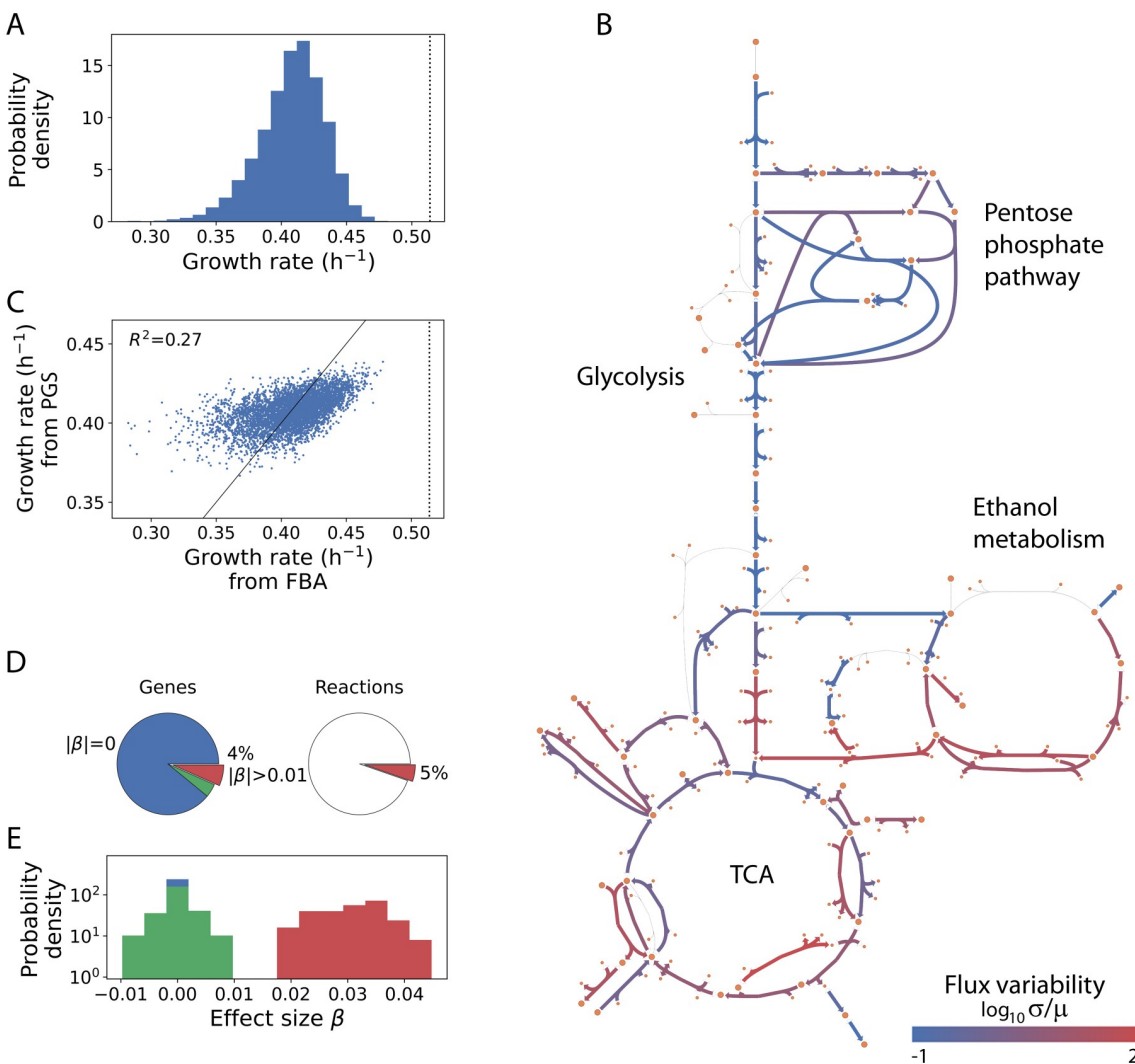

**Fig 2. The PGS reveals that a small number of genes account for 27% of growth variance in yeast metabolism.** A: We generated a synthetic population of $5 \times 10^3$ yeast metabolisms with relative gene (expression) dosages sampled from a normal distribution (Methods). For each individual, we calculate their growth rate with FBA. Such genetic variation induces a distribution of growth rates with mean and deviation $0.41 \pm 0.03$ h$^{-1}$. The vertical dotted line shows here the growth rate (0.51 h$^{-1}$) of a metabolism whose enzymes work at maximal efficiency. B: Scheme of central carbon metabolism showing population variation in each flux solution (coefficient of variation = $\sigma/\mu$, where $\sigma$ and $\mu$ is the population standard deviation and mean value, respectively, for each flux shown). C: We trained a PGS to predict growth rates from gene dosage data. The PGS explains 27% of the variation in the growth rate observed in the training population (obtained from FBA). Dotted line as in A. D: Effect size, $\beta$, for each gene in the PGS. Most of the genes, 88.7%, have null effect sizes (blue), while only 4.3% of them (red) are strongly associated with growth rate with $|\beta| > 0.01$ (the green color represents the part of non-zero weak effects). Genes with a strong effect control only 5% of metabolic reactions. E: Density distributions for each of the three categories above (the null class is a delta distribution, in blue, behind the weak effect distribution).

Consequently, we next hypothesize that the relevance of the genetic predictors derives from their direct contribution to the making of biomass precursors (Fig 3A). Fig 3B shows the mean aggregate metabolite production (or consumption, if negative) associated with all predictors in the population of metabolisms (Methods, S5 Fig). The strongest predictors only contribute significantly to a subset of precursors (11 out of 43), and in some cases, e.g., valine, lysine, etc., this represents the full production that is required for growth. Thus, the functional mode operating in the standard medium effectively selects a domain within the entire architecture of

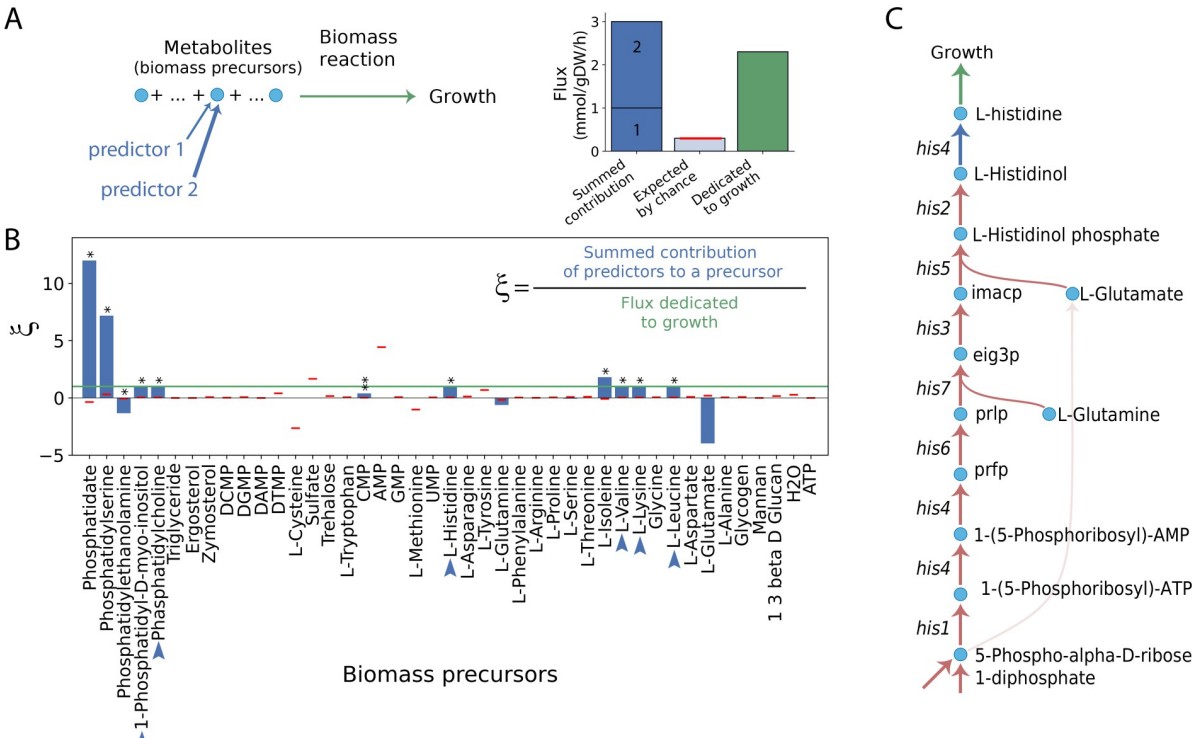

**Fig 3. Top predictor genes control the production of some metabolites required for growth.** A: The precursors in metabolic reconstructions are metabolites that ultimately fuel the biomass reaction, which simulates growth. We can calculate the contribution of those genes associated with the production of each specific precursor (e.g., predictors 1 and 2, the width of the arrow indicates their respective contribution) to compare it with an expected mean contribution (in this example, mean of $5 \times 10^3$ randomizations of two genes at a time, red horizontal line), and with the biomass reaction flux (green, this refers to the flux dedicated to growth). B: Population mean values of yeast metabolisms of $\xi$, the aggregate contribution of predictor genes to the production of each precursor relative to biomass consumption. In some cases (blue arrows), the sum of the contribution of various predictor genes produces 100% of the precursor consumed by growth (green line). As before, we tested for significant contributions after gene randomizations of $5 \times 10^3$ controlling for subset size (mean, red horizontal lines; *p< 0.05, **p< 0.01). The case of L-glutamate is not significant due to a large variation [23]. C: Part of the metabolic pathway that leads to the production of L-histidine. Although it is produced directly by *his4*, its production is influenced by upstream histidine-related genes, which also appear to be important predictors of growth.

growth. It is noteworthy that while the contribution to these factors is *direct* for only 9 genes – which are precisely producing these biomass precursors–, the rest of the predictors have a somewhat distributed effect on growth.

The case of histidine is exemplary (Fig 3C). Although it is produced only by *his4*, all histidine-related genes are crucial in providing intermediate metabolites and are therefore heavily involved in its overall production. This explains the case of *pmi40*, *sec53*, *dpm1* and *psa1*, which are predictor genes found upstream of mannan production, another important biomass precursor, while *pmt1–6* that finally produce said metabolite have null effect sizes. The same goes for *erg4* and sterol production.

## Pleiotropy does not inform the predictive character of a gene

The previous results underscore that the top predictors include genes that directly alter the availability of limiting biomass precursors and also genes whose impact comes from other upstream reactions. Could the systemic properties of metabolism capture this second aspect? We consider here first the pleiotropic character of a gene. One quantifies pleiotropy in

metabolic models as the number of biomass precursors whose maximum production is reduced by changing the dosage of a gene [24]. Therefore, the score includes system-wide phenomena such as metabolic compensation, rewiring, redundancy, etc.

Within highly pleiotropic genes, only some show large effect sizes in the PGS (Fig 4A). This result indicates that not all precursors incorporated into the pleiotropic score limit growth in the standard medium, i.e., not all are part of the functional mode that is operating. Indeed, we confirm that the identified predictors has a special impact on the production of a subset of precursors already identified in the previous section, namely some amino acids, phospholipids, mannans and sterols (Fig 4B). This reflects that pleiotropy fails to identify relevant genes for growth prediction, since it is an aggregate measure that includes the effect of a mutation in all biomass precursors, while only a few matter. In contrast, separating the individual contributions of the genes to each precursor results in a valid list of potentially growth-limiting metabolites.

## Growth predictors display either large additive or epistatic effects

A second systemic measure is epistasis (indicating gene-gene interactions). We intend to understand to what extent the variation of the predictors contributes additively to the phenotypic variation or if it also presents an epistatic component. To examine this, we present an approach based on *global* sensitivity analysis that allows us to precisely quantify the effect of

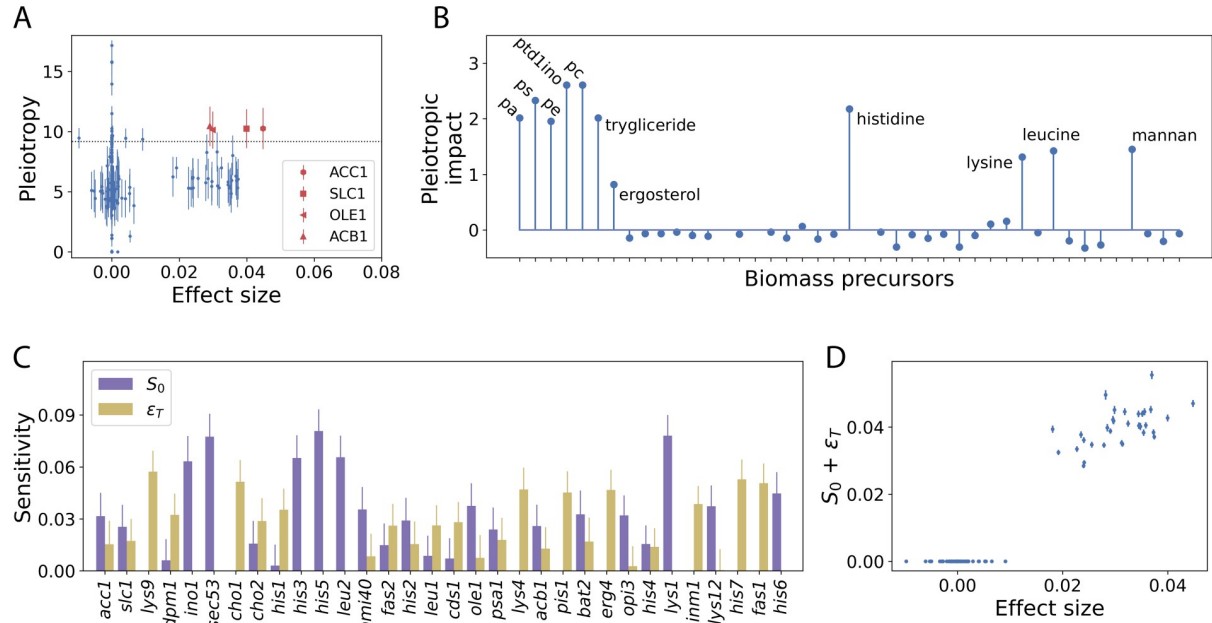

**Fig 4. System-wide effects of the top predictors.** A: Pleiotropy indicates the number of biomass precursors affected by a mutation of a given gene. Only a few top predictors have significant values of pleiotropy (above the 95th percentile of pleiotropy distribution for all genes, horizontal dotted line). This score is calculated considering a population of $10^3$ individual metabolisms. The graph shows the mean values (points) and one standard deviation (error bars) in this population. B: The pleiotropic impact quantifies how many of the genes with large effect sizes have an effect in the production of each biomass precursor. This is quantified by a z-score that compares the observed value with that expected by chance in a group of genes of the same size from the complete repertoire. We use abbreviations for phosphatidate (pa), phosphatidylcholine (pc), phosphatidylserine (ps), and phosphatidyl 1D myo inositol (ptd1yno). C: Global sensitivity analysis allows us to quantify both the additive impact of genes on growth rate, $S_0$ (purple), and the total epistatic effects, $\epsilon_T$ (yellow), which include second and higher order gene-gene interactions. Bars and vertical lines represent mean values and standard deviation, respectively, of $> 10^6$ simulations. D: The sum of the additive and epistatic effects correlates well with the polygenic score effect sizes ($\rho > 0.97$, Pearson) demonstrating the validity of the global sensitivity analysis. See Methods, for full details.

individual gene dosages on growth variation. Within this framework, two indices, $S_0$ and $\epsilon_T$, estimate these additive and non-additive contributions (*total order* epistasis), respectively (Methods and S1 Text).

Fig 4C shows that some predictor genes have large additive effects, $S_0$, and small total epistasis, $\epsilon_T$ (*ino1*, *his3*, . . .), while others show the opposite pattern (*cho1*, *his1*). It should be noted that the sum of all the effects, additive and epistatic, shows the maximum correlation with the effect sizes obtained in the PGS (Pearson's $\rho > 0.97$, Fig 4D and S6 Fig) confirming the validity of the overall sensitivity analysis. That the fraction of genes showing $S_0 > \epsilon_T$ and $S_0 < \epsilon_T$ is comparable highlights that the large effect sizes we obtain are associated with genes enriched by additive effects (something to be expected from a linear statistics formalism) but also with those with strong epistatic effects. This result confirms the view that the presence of epistasis inherent in the metabolic architecture (also known as functional epistasis; quantified here with $\epsilon_T$) may have a direct effect on the linear framework that captures the PGS [25–27].

Is there a structural basis for large $S_0$ or $\epsilon_T$? We investigated their relationship with several measures for each gene: the number of (active) reactions involved, the amount of flux they control, and the number of metabolites (precursors) they use. Among these, we find the largest correlations of $S_0$ (and $\epsilon_T$) with the number of reactions they control, $\rho = -0.19$ ($\rho = 0.19$), and the log of the summed absolute flux through their reactions, $\rho = 0.23$ ($\rho = -0.22$). Thus, one would expect the large *additive* effects to come from genes that regulate a small number of higher flux reactions. Conversely, genes that control a larger number of reactions with less flux result in larger *epistatic* effects.

## Predictability depends on genetic variation

Next, we investigate the impact on the predictability of the genetic variation available in the population, as measured by its standard deviation $\sigma_G$. We stress again that one advantage of our approach is that we can generate variation beyond what we might observe in a particular natural situation, where allele frequencies might be constrained by natural selection, genetic drift, etc.

We used ten populations with the same mean dosage and increasing $\sigma_G$ (Fig 5A and 5B; Methods) to calculate the corresponding growth rates with FBA and also to train a PGS in each case. We observe that the output of the PGS (coefficient of determination $R^2$) reaches a maximum optimal value for a given $\sigma_G$ (Fig 5B). This value results from stronger effect sizes for the same predictors found previously (Fig 2D). However, a better $R^2$ comes at the cost of a

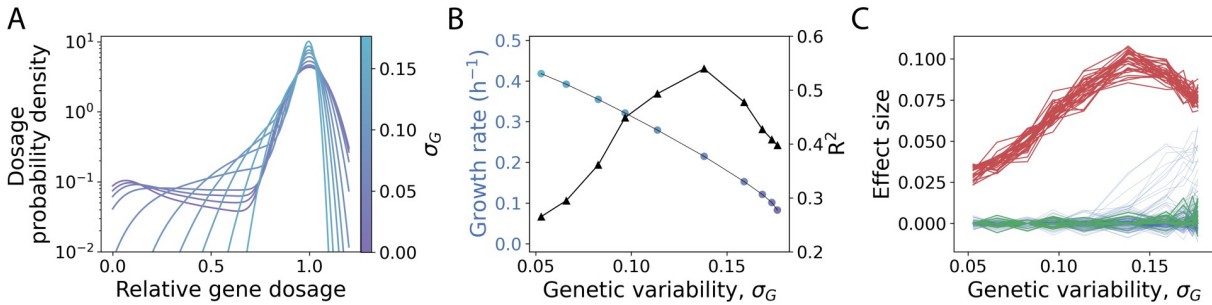

**Fig 5. Populations with different genetic variation reveal a common PGS architecture but also its predictive limit.** A: Probability density function estimate (kernel, bw = 0.3) of relative gene dosages of 10 populations with increasing genetic variation ($\sigma_G$). B: With FBA, we compute the growth rates (mean values on the left y-axis, colors correspond to different $\sigma_G$ as in panel A) and train a separate PGS for each population (coefficient of determination $R^2$ on the right y-axis, black triangles). C: Effect sizes of all genes as a function of $\sigma_G$. The lines are colored as in Fig 2E.

decrease in the average growth of the population, since the gradual increase in variation constrains the limiting fluxes more severely (colored circles, Fig 5B). In addition, the drop in PGS performance with greater genetic variation ($\sigma_G > 0.14$) is due to the accumulation of new predictors with a strong effect resulting from the appearance of new growth-limiting reactions (Fig 5C).

Therefore, even when the PGSs obtained with populations of different genetic variation include a considerable group of common predictors, these will differ by their strength. The individual prediction of a phenotype based on its specific set of alleles, in this case metabolic profile, will therefore depend on the genetic variation with which the PGS was obtained. In summary, these results reveal a trade-off between genetic variation, population fitness, and predictability. While it is desirable to increase the performance of a PGS by sampling from a population with high genetic variation, negative selection is likely to prevent optimal predictability scenarios [13]. Therefore, selection would pose severe limitations to the prediction of the phenotype.

### The prediction is portable across environments unless strong gene-environment interactions emerge

In our final study, we asked to what extent specific growing conditions influence predictive power. These conditions could alter the functional mode of the metabolisms, consequently modifying the PGS. Therefore, we randomly generate $>10^3$ (nutrients) environments of increasing richness (fixing for genetic variation as before, Methods). We then trained a separate PGS linked to growth in each medium (Fig 6A; we represent various conditions using flasks filled with differently colored media) to focus on the genes with the largest effect sizes ($|\beta| > 0.01$, as before). Effect sizes and predictability change with environmental richness (Fig 6B and 6C).

It could still be argued that these differences in predictability are due to subtle differences in metabolic solutions. Therefore, we control for environmental richness to quantify this. Differences in metabolic solutions are not correlated with predictability (Pearson's $\rho = 0.06$, using partial correlations to control for environmental richness), further pointing towards environmental richness as a valid measure that recapitulates the metabolic activity and similarity, anticipating predictability (Fig 6C).

We also observe that other genes appear recurrently as strong predictors in specific, generally poor environments (Fig 6B), and whose appearance leads to particularly strong PGS performance with up to $R^2 = 0.56$ (Fig 6C; genes listed in Fig 6D). Thus, while growth prediction is generally based on a core set of genes largely independent of the growth medium, strong gene-environment interactions can considerably improve the performance of a PGS.

Finally, the fact that the effect sizes change continuously with increasing richness (Fig 6B–6D) ensures the *portability* of a singular PGS (trained on a reference medium) to predict the growth rate of the same population in another medium of similar richness (Fig 6E). Fig 6F shows the performance of a PGS trained with data from the standard medium (PGS$_{std}$) to predict the growth rates obtained in different random media (as test populations). Indeed, we observe that beyond a certain environmental similarity (Methods) the portability of PGS$_{std}$ drops sharply along with the number of overlapping predictors between PGS$_{std}$ and PGS$_i$ (trained in the ith random environment). This enables us to distinguish high and low portability regimes.

### Discussion

The primary aim of our study is to investigate the relationship between the statistical associations observed in a PGS and the underlying GP map. Towards this goal, we developed an in-

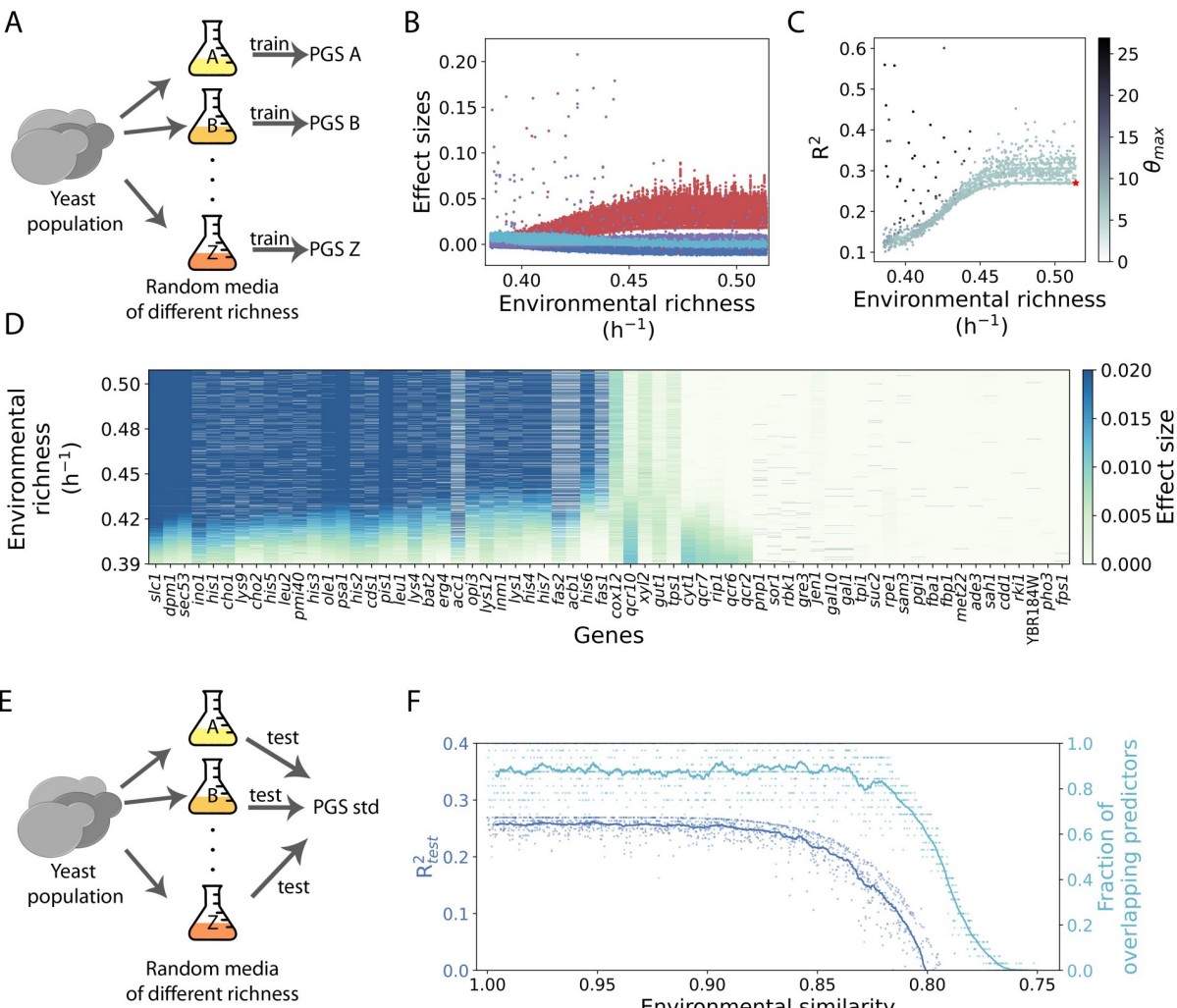

**Fig 6. Environmental effects on the predictability of the phenotype and the portability of a PGS.** A: We calculate the growth rates of a fixed population in $10^3$ random environments (represented with flasks of different colors) with different richness (Methods) to train a PGS in each of these. B: Effect sizes of all the predictors that make up the PGS as a function of environmental richness. We highlight previously identified top predictors (red, as in Fig 2), a new set recurring in poorer environments that are related to the mitochondrial respiratory chain (cyan), and genes that show large effects only in specific media (purple). C: The predictability of a PGS generally increases with environmental richness. However, in some media, predictability improves to $R^2 = 0.56$ due to strong gene-environment interactions identified by outliers in effect sizes ($\theta_{\max}$ is the z-score of the strongest predictor found in each PGS, given its constituent effect sizes). The red star denotes the standard medium case. D: The effect sizes of the genetic predictors follow a clear trend as a function of environmental richness. We explicitly show the values for all genes that have an effect size $\beta > 0.01$ in any PGS. E: Next, we tested the portability of the calculated PGS in the standard medium, $PGS_{std}$, that is, its ability to predict growth rates in different environments. F: The portability of $PGS_{std}$ (left y-axis) holds within a certain environmental similarity, measured as the ratio of random and standard medium richness. The drop in portability is related to the decreased overlap of predictors between $PGS_{std}$ and the corresponding PGS of the medium (right y-axis). Dots and lines correspond to individual means and a running average ($n = 50$), respectively.

silico framework to generate genetic and phenotypic variation in a population of yeast metabolisms. The computational model acts as an explicit GP map in which the quantitative interpretation of the GRRs incorporates a fundamental structural layer [21]. This feature and the fact that the genetic variation produced is not restricted as in the case of natural populations allows us to study in depth how and why a PGS can predict the phenotype (growth rate).

Computing a case study, we obtain an $R^2 = 0.27$ and 32 genetic predictors with large effect sizes (Fig 2D and 2E). Which genes act as predictors result from the combination of two

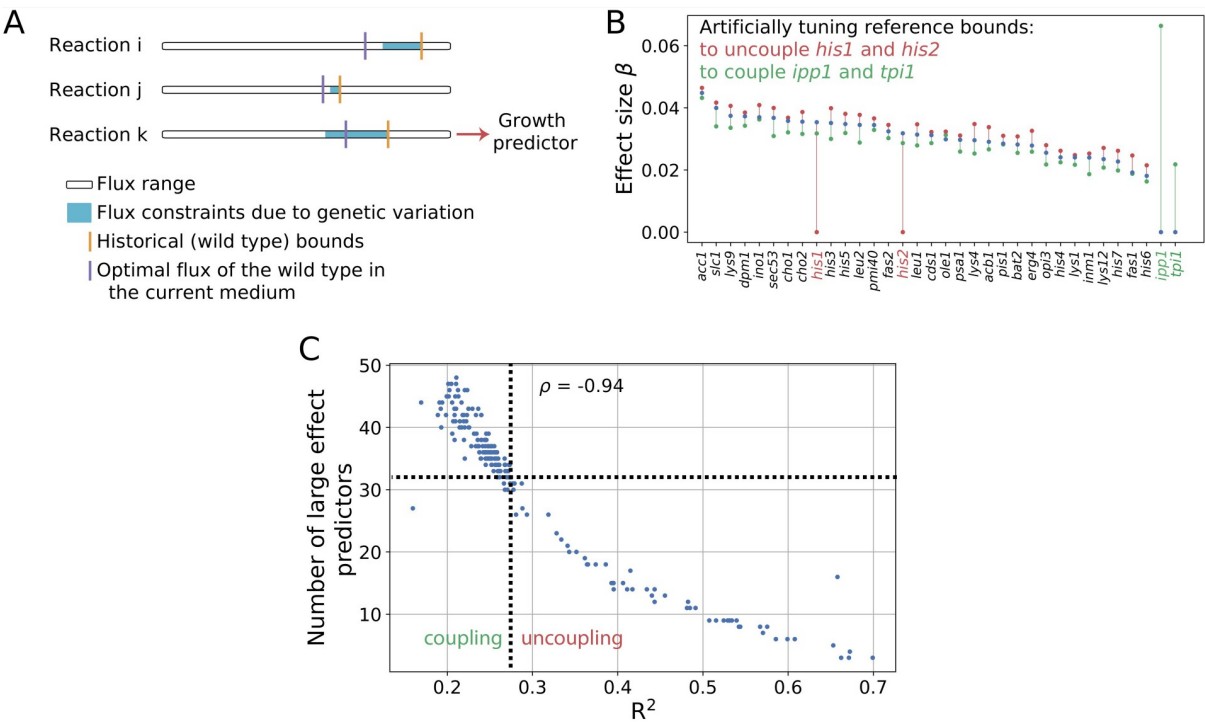

**Fig 7. Functional mode and historical bounds determine the predictive character of a gene in metabolism.** A: The range of flux potentially available in three reactions is represented by the white bars (from zero on the left to a given value on the right). Within this range, the purple line indicates the optimal flux in a given growth medium (functional mode), while the orange line indicates the wild-type reference (historical bound). Genetic variation in a population translates into a range of flow restrictions (cyan) around this bound. Variation can become cryptic if it does not restrict the flux below the optimal value (for example, the reactions *i* and *j*). This may be a consequence of the gene reaction rules (S1 Fig). When genetic variation limits the flux of the functional mode, its associated genes can become predictors of growth (as in the *k* reaction). B: List of predictors and effect sizes for the original PGS before (blue) and after (red and green) artificially manipulating changing the reference bounds. By increasing them in the reactions associated with *his1* and *his2* or decreasing those related to *ipp1* and *tpi1*, these genes stop being (red) or become (green) growth predictors (effect on PGS changes to/from zero, respectively). C: If we iterate the above experiment and uncouple (red) or couple (green) randomly selected groups of predictor genes (linked to between 10 and 60 reactions), we find that the performance of a PGS is negatively correlated with the number of large-effect predictor genes (linear $\rho = -0.94$). Therefore, the more growth-limiting reactions, the more predictors and the more deteriorated the predictability (lower $R^2$). The dotted lines indicate the original situation.

factors: the quantitative flux *required* in the reaction(s) associated with the predictor gene and the flux constraints *derived* from the corresponding genetic variation (Fig 7A illustrates this combination). The former contributes to the functional mode of metabolism, which incorporates environmental information (growing conditions). The latter integrates information on allele frequencies in the population with the inherent architecture of metabolism and its reference bounds, a product of the environmental and genetic conditions experienced during its evolutionary history.

Notably, both GRRs and the reference bounds can act as a sieve of genetic variation causing part of it to be cryptic [28–30]. For instance, GRRs could describe the presence of an isozyme, and this redundancy would prevent such gene from becoming a phenotypic predictor. Genetic variation would also be cryptic when the reference bounds substantially differ from the required flux in a given condition (Fig 7A). This again silences the functional impact of the genetic variation in the population.

The presence of the specific predictors thus depends on the combination of the working regime, i.e., functional mode, and the evolutionary history of the metabolism. By manipulating reference fluxes, we confirmed this reasoning (Fig 7B). This explains the specific cases of

mannan and sterol production: upstream genes are growth-limiting because they are working close to their production limit in opposition to other downstream enzymes. Note, however, that growth predictors should be necessarily associated with biomass precursors, either directly or in upstream reactions, since the biomass reaction represents the genetic architecture of the trait of interest, growth (Fig 3A and 3B).

This is the reason why pleiotropy, as *global score* of the impact of mutations in all biomass precursors, is a poor measure of the predictive character of a gene. However, the disaggregated information partially reveals the composition of the biomass reaction (Fig 4B). Thus, the PGS provides a solid but partial understanding of the trait architecture, exclusively the domain associated with the precursors needed in a particular working regime. This dependence of the genetic architecture underlying a complex trait on activation conditions could occur in most PGS scenarios.

We reiterate that the primary aim of our study is to investigate the metabolic aspects that underlie the mechanistic relationship between a phenotype and the statistical prediction methods, specifically focusing on deriving a biological reasoning behind the precise value of $R^2$ of a polygenic score. The general claims and conclusions of our study do not rely on the specificities of the model, including the metabolic network itself (Methods), the flux solutions or the true growth prediction ability in specific experimental conditions. By examining the metabolic models and understanding the biological mechanisms driving growth rate predictions, we provide a comprehensive and mechanistic interpretation of the statistical predictions. This approach allows us to establish a solid biological foundation for the predictive performance of PGS and deepen our understanding of the genotype-phenotype relationship in the context of metabolism.

## GP map nonlinearities

Those genes whose dosages eventually limit growth induce nonlinearities in the metabolic GP map. On this basis, each individual metabolism is a case where only one, or a few, dosages are particularly limiting, and the exact dosage vary between them (S7 Fig). But the combination of these individuals in a population generates functional cross-dependencies, increasing the number of rate-limiting enzymes and, consequently, reducing the predictability (the greater the number of predictors, the weaker the prediction, $\rho = -0.94$ linear, Fig 7C). This also describes the beneficial effect of sufficiently large gene-environment interactions on predictability producing fewer predictors and, therefore, better predictability (Fig 6C).

Moreover, global sensitivity analysis confirms that predictors associate with functional epistatic effects (Fig 4C). Even so, effect sizes capture the sum of both additive and epistatic contributions (Fig 4D). Indeed, a PGS should capture nonlinearities, since the minimization of error due to a linear regression incorporates all data points, including those coupled to nonlinear regimes. Still, and despite epistatic effects, we notice that the sum of additive terms accounts for over 75% of the total phenotype variation ($\sum S_0 > 0.75$, Fig 4C). A signal that was also observed in a recent analysis of growth traits using a cross between two yeast strains [31]. How is this additive component generated in our case? This is conditional on the structure of the GP map, in particular on the monotonic relation between gene content and phenotype (order preservation of its responses to gene dosage, S7 Fig) [32]. This monotonicity –particularly strong in the case of metabolic GP maps– is "broken" by the aforementioned functional cross-dependencies (see extended discussion on additivity of metabolic GP maps in S1 Text and S8 Fig).

## Prediction portability

Populations experiencing an intermediate genetic variation correspond to maximum predictability, in line with previous results on extreme allele frequencies [33]. Such an

increase in $R^2$ comes at the cost of population growth (Fig 5B). Therefore, this maximum could be unattainable due to negative selection [13]. This limitation will be less apparent in those GP maps in which genetic variation is less likely to cause loss of function, as in our case, and more forms of gain of function are possible. In addition, the list of genetic predictors remains largely the same, independent of the amount of genetic variation in the training population (Fig 5C). This is also observed when one determines predictors associated with populations growing in diverse media of similar richness, with two consequences (Fig 6D). First, these results ensure the portability of the PGSs. Second, and as stated before, the predictors are specific to the evolutionary history of the metabolism. As a consequence, portability might not necessarily hold for more idiosyncratic, and maybe more realistic, histories.

## Conclusions

The purpose of our study is to provide a mechanistic interpretation of the predictions derived from a PGS. Our results illustrate how the intrinsic GP map structure and the population allele distribution combine to determine phenotypic prediction and how the environment influences this combination. We presented this explicitly as a synergy of three components –functional mode, evolutionary history, and phenotypic architecture—benefiting from a computational framework that enabled us to examine the problem beyond current experimental setups. While the principles learned here are general in the sense that any metabolic GP map will display them, we still have a lot to learn about those associated with many other complex traits. Integrating explanation and prediction guarantees many stimulating future discussions.

## Materials and methods

### Metabolic models

Whole-genome metabolic models integrate the stoichiometry of the metabolic reactions of a particular organism [22]. With the use of these models, FBA is a computational method to simulate the function of metabolism by computing an optimal network solution given an objective function where fluxes are stable (see S1 Text for further description). Specifically, we used parsimonious FBA, which also minimizes the absolute fluxes within a metabolism [34]. We focus on the prediction of biomass production, an analogue of growth rate. We used the genome-scale metabolic reconstruction of *Saccharomyces cerevisiae* iND750 [35] together with the Cobra toolbox for Python [36] to compute the growth rate of numerous genetic variants in either a standard medium or random media and the Escher package to depict the central carbon metabolism [37]. This choice of model tries to balance the presence of sufficient biological details with accessible computational time. Still, our results and the mechanisms underlying phenotype prediction are robust when using the most recent yeast metabolic reconstructions iMM904 and yeast8 in equivalent standard media (PGSs obtained with these models lead to $R^2$ = 0.18 and $R^2$ = 0.17, respectively).

With respect to the examination of the association between predictors and metabolic subsystems, e.g., glycolysis (S3 Fig), we made use of the model information on which metabolic subsystems are assigned to the metabolic reactions. We impute a specific subsystem to a gene only if all reactions in which it participates belong to the same subsystem. Among all 750 genes present in the model, a subset of 42 had either none or multiple subsystems associated. We did not consider this subset when we compute S3 Fig.

## Quantitative mutations

We imagine that genetic variation at the nucleotide level causes a corresponding reduction in the dosage of gene expression, or equivalently in the efficiency of the enzyme, which ultimately impacts the associated fluxes. This procedure incorporates two steps (S1 Fig). First we need to calculate the wild-type "reference" limits of each reaction flux. We then calculated how the reduction in gene expression dose translates into a reduction in flux (via its associated reactions) relative to these earlier references. This second step also requires the quantitative interpretation of GRRs, also known as gene protein reactions.

`Step 1`: To obtain the reference fluxes, we calculated $2 \times 10^4$ optimal solutions of metabolisms exposed to random environments (see section on growth media) and random genetic backgrounds (by random sampling of flux boundaries from a distribution uniform in the range [0.100] mmol/gDW/h). Therefore, we derive $2 \times 10^4$ flux values for each reaction for which the reference flux is the corresponding maximum (and minimum if reversible) of these values. This collection of situations is what we consider the "evolutionary history" of yeast metabolism.

`Step 2`: To find how reducing the dose of an enzyme translates into reduced flux through its reactions relative to reference, we quantitatively interpret GRRs. This is necessary because some reactions may require several subunits or only one of several isoenzymes. GRRs can contain operators `AND` and `OR` that act on pairs of genes, we consider them equivalent to `min` and `sum`, respectively, that act on relative dosages of genes. This approach is similar to those used in noise propagation [38], or by the Escher package [37]. In all cases, the upper/lower limits are always calculated and set according to the reversibility of the reactions, while the limits of ATP maintenance, biomass production, and exchange reactions remain unchanged.

Note that this protocol is comparable to a previous approach in which genetic variation was also mapped to flux constraints [21]. The authors build the allele–to–flux constraint map coupled with the performance of a novel objective function to classify antibiotic resistance in a fixed medium. Our approach, however, assumes flow restrictions imposed by a history of past genetic and environmental adaptations, and in this sense is more comprehensive, incorporating explicit information about which genes are involved in exact reactions (GRRs). Finally, this method of associating genotypic variation with flux variation, common in other previous work in the context of metabolism, e.g., [39], is a possible approach to the problem of how to characterize the genotype-to-parameter map for any model [17]. The study of this map is a very exciting research program in itself, but it is beyond the scope of this paper.

## Genetic variation

We generate genetic variation by sampling gene (expression) dosages from a probability distribution. Unless otherwise stated, we use a normal distribution with unit mean and standard deviation $\sigma = 0.1$. In this way, $\sigma$ directly reflects the variation in gene dosages in the population. This distribution is derived from Fisher's original infinitesimal model or from the Gaussian Descendants derivation, where different levels of parenthood result in different $\sigma$ under neutral evolution [40, 41]. This procedure generates populations that are in linkage equilibrium. Note that, in this context, a *wild-type genotype* has all gene dosages equal to unity (vertical dotted line in Fig 2A, main text).

We also designed genetic variation based on gamma distributions with shape parameters $0.5 < k < 200$ and scaled so that all distributions had the same mean. It is important to note that gene dosages greater than 1 are not beneficial, as the wild-type limits are the extreme values observed (statistically). Therefore, to avoid including additional cryptic genetic variation,

gene dosages greater than one were trimmed by unity. Thus, the resulting genetic variation in our standard population is $\sigma_G = 0.05$ (Fig 5, main text).

## Growth media and environmental variability

Broadly, we distinguish three types of media: a minimal medium, a standard medium and randomly generated media. The minimal medium consists of the most fundamental components strictly necessary to support growth, albeit when supplemented with few other nutrients, it is defined by unlimited import and export of $H_2O$, $CO_2$, ammonia, phosphate, sulfate, sodium and potassium and is aerobic with an import rate of 2 mmol/gDW/h of $O_2$. We avoid incorporating fermentative phenotypes into our growth prediction model as they can limit accuracy and introduce potential artifacts. The standard medium is additionally composed of a glucose import rate of 20 mmol/gDW/h [30, 35]. Random environments are generated following a previous protocol [42]. Briefly, we supplement the minimal medium with an additional number of components, comprising all possible carbon and nitrogen sources, such that the probability of including any follows an exponential distribution with mean $m = 0.10$, with other values producing similar results. Then, for each component, we obtain their maximum import rates from a uniform distribution between 0 and 20 mmol/gDW/h.

We define the richness of a medium as the growth rate of the wild-type genotype (see definition above) and the environmental similarity between two media as the ratio of their richness. Metabolic models can support arbitrarily large growth rates, thus to avoid randomly generating unrealistically rich media we consider those with richness less than or equal to that of the standard medium. In addition, we discarded media that support biomass production rates <70% of those of the standard medium to avoid possible natural and model artifacts related to our implementation of quantitative mutations (see Results). This is an alternative approach to Constrained Allocation FBA (CAFBA), which limits the growth rate of metabolic models based on resource allocation principles by fixing a medium and tuning a parameter related to proteome fractioning [43].

## Polygenic score

We used a high-dimensional regression framework for polygenic modeling and prediction:

$$\overrightarrow{y}_{N\times1} = \mathbf{G}_{N\times M}\,\overrightarrow{\beta}_{M\times1} + \overrightarrow{\epsilon}_{N\times1} \tag{1}$$

where $N$ is the sample size, $M$ is the number of genes, $\overrightarrow{y}$ is the vector of phenotypes (growth rate in 1/h), $\mathbf{G}$ is the genotype matrix, $\overrightarrow{\beta}$ is the vector of effect sizes of the genes (in 1/h), and $\overrightarrow{\epsilon}$ is some noise assumed normal with unknown variance. The generated data was fit using Least Absolute Shrinkage and Selection Operator (LASSO) a type of regression that under bayesian statistics assumes prior Laplace distributions in each coefficient, instead of uniform distributions as in the case of Ordinary Least Squares. Consequently, with LASSO some parameters are automatically zero [44], hence making it a remarkable alternative to pruning and thresholding (P+T) or other regularization methods [2, 45]. In addition, we compute the best value of the shrinkage parameter with five-fold cross validation. That effect sizes show a bimodal distribution makes our results robust to the application of other regularization, or feature selection methods (Fig 2E). This situation differs from those observed in association studies where, typically, the number of predictors is several orders of magnitude larger than the size of the training population posing a few limitations that we overcome with our in-silico approach. Moreover, we avoid sample population biases by deliberately imposing genetic variation on all genes present in the model. Therefore, by training the PGS with an engineered

population we can easily overcome data shortcomings commonly found in experimental association studies.

We additionally tested other fitting methods, for which we obtained similar mean $R^2$ (and one standard deviation) values under a 5-fold cross validation of the training population: OLS ($R^2 = 0.19 \pm 0.01$), LASSO ($R^2 = 0.25 \pm 0.01$), Ridge ($R^2 = 0.21 \pm 0.01$) and Elastic net ($R^2 = 0.25 \pm 0.01$). All major conclusions are qualitatively equivalent as the observed bimodal distribution remains.

To evaluate possible overfitting in our training protocol, we systematically generated $10^3$ independent test data of $10^4$ individuals each), and with the same size and mutational distribution as the training population. For each test set, we computed the $R^2_{\text{test}}$ to obtain a mean $\langle R^2_{\text{test}} \rangle = 0.24 \pm 0.01$ across all sets. This reveals that the computed PGS only slightly overfits the training data, and that we should expect a small loss of predictability when applied to different test populations ($R^2 > \langle R^2_{\text{test}} \rangle$ with $p < 4 \times 10^{-3}$).

## Global sensitivity analysis and total epistasis

Global sensitivity analysis breaks down the variation in a model's output into different terms when all variables fluctuate simultaneously. We use the method first proposed by Sobol for its easy implementation and interpretation [46, 47]. Note that this differs from previous flux-based applications [48, 49]. Briefly, we focus on two indices for the *ith* gene, the first order index $S^i_0$ and the total effect index $S^i_T$. The former quantifies the additive part of the variation associated with a gene while the latter quantifies its total contribution, additive and all non-additive effects. From these, we derive the total epistasis that explains all, and only, non-additive effects like $\epsilon^i_T = S^i_T - S^i_0$ and its error $(\Delta \epsilon^i_T)^2 = (\Delta S^i_0)^2 + (\Delta S^i_T)^2$.

We computed all indices and their errors with Monte Carlo estimators using over $10^6$ samples [47, 50]. We carry out these calculations with pairs of genotypes sampled from the original population growing on standard medium. A detailed description of the protocol and the equations are available in the S1 Text. Finally, note that we do not show negative values for $S_0$ and $\epsilon_T$ as they are unrealistic and should be considered null according to their error bounds.

## Pleiotropy

In metabolic models, the pleiotropy of a mutation is generally computed as the number of biomass precursors whose maximum production is limited by the mutation, following a previous protocol [24, 51]. In short, we simulated the excretion of a given metabolite by adding an exchange reaction to the model and maximizing the flux through this reaction. Then we consider that a gene limits the production of a metabolite if, when knocked-down by 90%, its excretion rate decreases. As pleiotropy is strongly dependent on the genetic context, we computed the mean value across $10^3$ individuals of the population due to the large computational load. We used a 90% decrease in dosage to avoid artifacts derived from gene essentiality, but our results are robust when using other values.

## Supporting information

**S1 Text. Supplementary note.** Supplementary note including a brief introduction to flux balance analysis, global sensitivity analysis, a detailed discussion on the additivity in genotype to phenotype metabolic maps and the GO enrichment results of predictors with the largest effect sizes.
(PDF)

**S1 Fig. Modeling of quantitative mutations.** We characterize mutations by a decrease in enzyme efficiency with respect to a wild-type "reference", or "maximum" value. A: To find the wild-type lower and upper bounds of each reaction $r$, $f_r^{lb}$ and $f_r^{ub}$, we expose the yeast metabolism to a series of environmental and genetic conditions and compute the minimum and maximum fluxes observed in the solutions taking into account their reversibility. Specifically, we compute pairs of optimal solutions in $10^4$ random media from a totally unbounded and a randomly bounded yeast metabolisms (random bounds change in every medium; Methods). The bound for a given mutant and reaction is the product of the corresponding wild-type bound and a fractional value resulting from the quantitative interpretation of the associated gene reaction rule. B: Here we show a detailed example involving the malate synthase reaction MALSp which is mediated by two isozymes dal7 and mls1. The mutant upper bound, $\tilde{f}_{\text{MALSp}}^{ub}$ where ˜ denotes mutant, is the product of the wild-type upper bound $f_{\text{MALSp}}^{ub}$ and the result of the gene reaction rule $\delta_{\text{MALSp}}$. C: The gene reaction rule in the case of MALSp reads "dal7 or mls1", so we compute the sum of the relative gene dosages in the mutant of enzymes dal7 and mls1, $\tilde{g}_{dal7}$ and $\tilde{g}_{mls1}$ respectively.
(TIF)

**S2 Fig. Genetic variation leads to flux variability, which is well described by the mean metabolism.** A: Fluxes that are accessible in the population, i.e. maximal bounds (vertical black lines) and range of values observed in the population (vertical red lines) for each reaction (x-axis). Blue lines represent 70% of such maximal bounds, which is approximately the largest restriction in the default population (with dosages sampled from a normal distribution with unit mean and $\sigma = 0.1$). We find that the genetic variation with which the population was generated leads to variability in some solution fluxes of the individuals, which ultimately translate into growth variability. B: Despite this variability in solution fluxes, we can define a "mean" metabolism in which the flux through each reaction is the observed mean across the population. Black dots depict data of the reactions of all individuals in the population, and the inset shows the distribution of linear correlations between each individual's solution and the mean metabolism.
(TIF)

**S3 Fig. Genetic predictors belong to a handful of metabolic subsystems.** A: Manhattan-like plot showing the effect sizes (y-axis) of genes grouped by yeast metabolic subsystems (x-axis; arbitrary colors). We find that genes with large effect sizes belong to a handful of subsystems related to protein synthesis, cell membrane and organelle compartmentalization. B: Effect sizes of all predictors identified in the PGS (colors as in Fig 2) and their corresponding metabolic subsystem.
(TIF)

**S4 Fig. Biomass precursors and their stoichiometric coefficients in the biomass reaction.** The biomass reaction involves 43 precursor metabolites (x-axis) but with stoichiometric coefficients spanning several orders of magnitude (y-axis, in log scale). For example, the most consumed precursors are ATP and water.
(TIF)

**S5 Fig. Contribution of genetic predictors to the mean production or depletion of biomass precursors.** Each genetic predictor (x-axis; sorted by effect size) participates in a number of reactions that might involve biomass precursors (y-axis). We here show the mean consumption (red circles) or production (blue circles) across the entire population ($10^4$ individuals). Circle sizes are proportional to the absolute value of the mean contribution relative to the

biomass consumption.
(TIF)

**S6 Fig. Effect sizes correlate with global sensitivity indices.** A: first order index $S_0$, B: total epistasis $\epsilon_T$ and C: total effects $S_0 + \epsilon_T$ as a function of effect size (Methods). The linear correlations among all genes are $\rho_{S_0}^{all} = 0.63$, $\rho_{\epsilon_T}^{all} = 0.48$ and $\rho_+^{all} = 0.98$ respectively; or among only large effect predictors $\rho_{S_0}^{pred} = 0.19$, $\rho_{\epsilon_T}^{pred} = -0.08$ and $\rho_+^{pred} = 0.57$. We show the mean values and a standard deviation of $> 10^6$ simulations for each gene (Methods).
(TIF)

**S7 Fig. Dosage-response profiles of all predictor genes.** We computed the dosage-response profiles for all predictor genes in 200 genetic backgrounds by individually tuning the corresponding dosage from $g = 0$ to $g = 1$ and computing the growth rate with FBA. Observe that i) all top predictors (with $\beta > 0.01$) are essential. That is, growth is null if $g = 0$; ii) that only top predictors display a recurrent dosage-response profile and that iii) the profiles of genes with $\beta < 0.01$ are constant in the range mostly accessed by the population $0.7 < g < 1$.
(TIF)

**S8 Fig. Additivity of metabolic GP maps.** Growth costs (colorbar) caused by individual virtual mutations of enzymes (rows) for $10^3$ different individuals (columns) identify the structure of limiting reactions. The figure displays two populations for which the PGS performance differs ($R^2 = 0.26$ and $R^2 = 0.84$ in A and B, respectively). By clustering individual patterns, we recognize that a more straightforward structure in the dendrogram leads to better prediction (panel B with larger $R^2$ than in panel A with smaller $R^2$).
(TIF)

## Author Contributions

**Conceptualization:** Pablo Yubero, Alvar A. Lavin, Juan F. Poyatos.

**Formal analysis:** Pablo Yubero.

**Funding acquisition:** Juan F. Poyatos.

**Investigation:** Pablo Yubero, Alvar A. Lavin, Juan F. Poyatos.

**Methodology:** Pablo Yubero.

**Software:** Pablo Yubero.

**Supervision:** Juan F. Poyatos.

**Writing – original draft:** Pablo Yubero, Juan F. Poyatos.

**Writing – review & editing:** Pablo Yubero, Alvar A. Lavin, Juan F. Poyatos.

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
