## [Decision Letter · Decision Letter 0]

6 Jul 2023

Dear Dr. Poyatos,

Thank you very much for submitting your manuscript "The limitations of phenotype prediction in metabolism" (PCOMPBIOL-D-23-00225) for consideration at PLOS Computational Biology. As with all papers peer reviewed by the journal, your manuscript was reviewed by members of the editorial board and by several independent peer reviewers. Based on the reports, we regret to inform you that we will not be pursuing this manuscript for publication at PLOS Computational Biology.

Dear Dr. Poyatos,

After finally finding sufficient reviewers for the manuscript who could consider it carefully, the reviewers found the manuscript interesting and novel, but found the study in need of further development. Some issues are simply semantics, but some issues are more substantive. Unfortunately, at this point we are not able to accept the study as it is.

The reviews are attached below this email, and we hope you will find them helpful if you decide to revise the manuscript for submission elsewhere. We are sorry that we cannot be more positive on this occasion. We very much appreciate your wish to present your work in one of PLOS's Open Access publications. 

Thank you for your support, and we hope that you will consider PLOS Computational Biology for other submissions in the future.

Sincerely,

William Cannon

Guest Editor

PLOS Computational Biology

Mark Alber

Section Editor

PLOS Computational Biology

Reviewer's Responses to Questions

**Comments to the Authors: **

Reviewer #1: In this manuscript, the Authors use a genome-scale metabolic model of yeast in order to establish a polygenic scoring metric. The Authors apply randomization to generate their datasets, and claim to have used their polygenic score to identify genes, which influence the read-out (predicted specific growth rate) in a dose-dependent manner. I believe the study suffers from fundamental flaws which all should be addressed preceding publication in any scientific journal.

Major comments

Many of the points listed below will point into the main issue of the study as I see it: application of statistical methods without addressing biological considerations.

To remove bias, linked to specific growth environment(s), Authors use some randomization-based pipeline to generate, as they call them, environments “representing evolutionary history” of yeast. Judging by the description in the Methods, these environments are essentially a panel of supplements to the very same medium. 

I do not treat this as a valid representation of the potential environments of S. cerevisiae – there are plenty of carbon and nitrogen sources these cells can consume, and the Authors only use glucose as a primary and very abundant (20 mmol/gDW/h, around the maximal uptake rate S. cerevisiae ever show) carbon source. What about the dozen(s) of other carbon sources S. cerevisiae can (co)consume? Moreover, the Authors try to avoid generating “rich” media, which by itself breaks down their whole idea of comprehensive representation of different natural environments.

More to the point of defining the growth environments. Inorganic salts are allowed to take up ad libitum, glucose – pretty much as well, but why oxygen uptake is the only hard flux constraint here? Intuitively, this “locks” the phenotype space to only include fermentative phenotypes, which might be not at all reasonable when, e.g. tested mutants exhibit low specific growth rate. Moreover, the numbers for uptake bounds themselves are not explained (no references or other support for the values of 2 and 20 mmol/gDW/h for oxygen and glucose, respectively)

Eventually, the Authors construct a polygenic score for predicting growth rates, and it does not perform so well (R2 < 0.3). My first feeling why the R2 score is low is the following: based on the description of the routines, is that the Authors do not address the issue stemming from the presence of isozymes – in the metabolic model, preventing the flux that uses a (biologically known) major isozyme will have little-to-no effect if there exist alternative isozymes and/or cofactors that the enzymes use. Authors briefly touch upon this in the Discussion (Lines 254-255) but there are no clear signals this is resolved in the current study.

The major insight of the manuscript (Section “Few metabolic functions limit growth”) is, in my opinion, trivial. 

Line 117-118: “However, we only found a few predictor-enriched metabolic subsystems (Methods, Fig. S3 and Table S1). These subsystems specifically involve the production of 118 biomass precursors.”. Naturally this conclusion arises when considering that ALL of these precursors must be made de novo or imported in order to produce biomass, which is the ONLY read-out that Authors use. The point that Authors consider minimal media only does not help here either – a single knockout in a single pathway of making one of the biomass precursors could prevent biomass formation in minimal but not rich media. 

Minor comments

I find the Author’s interpretation of Gene-Protein-Reaction (GPR) associations strange (Lines 80-82): “Subsequently, the dosages are quantitatively interpreted in the model by Gene Reaction Rules (GRR): Boolean relationships between enzymes that define which (and how) they participate in the reactions (Fig. 1B, Methods and Fig. S1).” I wonder how Boolean descriptions are supposed to be quantitative, in the meantime capable of acquiring only values of “true” and “false”. The Authors claim to give quantitative meaning through applying flux constraints, but this has NOTHING to do with the GPR associations themselves!

I do not completely follow why the Authors are exploring these artificial variability/mutation profiles without first consulting a profound panel of natural genetical variability in the same organism, S. cerevisiae, in different ecological niches (Peter 2018 Nature). 

Moreover, fluxomics studies of multiple mutants and growth in different conditions (E. coli and S. cerevisiae) are available – why not to make good use of that data?

Reviewer #2: Summary: 

The authors set out to utilize a genome-scale metabolic model to facilitate prediction of metabolic phenotypes in a controlled gene/environment setting. Perhaps the most interesting aspect of the study is the interaction between genes and environments which has the potential to be of interest to a general scientific audience. Overall, the present study is interesting but requires clarifications and additional data to warrant publication.

Major Comments:

General question – how does this model consider the established feedback mechanisms that exist in yeast to induce biosynthetic machinery under conditions of limited availability of biomass precursors? The most relevant example is histidine. The lack of histidine can be sensed by GCN2, and biosynthetic machinery transcriptionally activated by GCN4. Is there a way in which this type of regulation can be included in the model?

How do the authors explain the case of mannan production. The text states that upstream genes in the pathway are predictors, however, the final enzyme responsible for production of the terminal metabolite “have null effect sizes”. Does this imply that the production of the terminal metabolite is not causative for the growth effect? If so, do the preceding metabolites have alternative metabolic routes? Could the production/consumption of co-factors be important (NAD+, etc)? The same information is desired for erg4 and sterol production. 

Can the authors provide additional details on the effect of genetic variation on the growth of the population? 

The authors should include some metrics for predictor genes across experimental conditions – for example his4 appears to have been identified several times. “We also observe that other genes appear recurrently as strong predictors in specific, generally poor environments (Fig. 6B), and whose appearance leads to particularly strong PGS performance with up to R2 = 0.56 (Fig. 6C).” It might be useful to know what these genes are, or explicitly mention that those are the genes shown in Fig. 6D.

The authors utilize a LASSO regression for generation of PGS and identification of genes with a large beta. In the methods, the authors state “That effect sizes show a bimodal distribution makes our results robust to the application of other regularization, or feature selection methods (Fig. 2E)”. It would be interesting to test if other regression models (perhaps ridge or elastic net) yield similar number of genes with large beta.

Minor Comments:

Some mention should be made about how the model handles multi-gene protein complexes which require each protein to function. Does altering gene dosage of a catalytic subunit reflect a growth/biomass defect for other components of the complex?

Line 129 – “especial” to “special”

Reviewer #3: in the attachment

**Have the authors made all data and (if applicable) computational code underlying the findings in their manuscript fully available?**

Reviewer #1: Yes

Reviewer #2: Yes

Reviewer #3: Yes

PLOS authors have the option to publish the peer review history of their article (what does this mean?). If published, this will include your full peer review and any attached files.

Reviewer #1: No

Reviewer #2: No

Reviewer #3: No

---

## [Editor Report · Decision Letter 1]

9 Sep 2023

Dear Dr. Poyatos,

I am returning the manuscript to you for revision before I send it out for review. While I really like the intent of the study, your intent is not coming across as it is written (hence, the reviewer comments). The revisions that you made in this last round are solid clarifying points, but reviewers are getting lost well before the points in the main discussion. In the response to reviewers, you clearly state that "The primary objective of our study is to investigate the metabolic aspects that underlie the mechanistic relationship between a phenotype and statistical prediction methods, with a specific focus on understanding the biological reasoning behind the precise value of R2 for a polygenic risk score (PGS)." Yet this intent does not come across in the Introduction. For demonstration, I have taken the liberty to revise the final paragraph of your introduction with the hope that it is clear what the intent of the study is. Elsewhere I have also made some changes to help orient readers. These are all marked in red in the pdf that I am attaching. Please take a look at these, and if I have captured the intent of the study, please consider revising your manuscript similarly (but in your own words). 

[1] A letter containing a detailed list of your (previous) responses to the review comments and a description of the changes you have made in the manuscript. Please note while forming your response, if your article is accepted, you may have the opportunity to make the peer review history publicly available. The record will include editor decision letters (with reviews) and your responses to reviewer comments. If eligible, we will contact you to opt in or out.

Sincerely,

William Cannon

Guest Editor

PLOS Computational Biology

Mark Alber

Section Editor

PLOS Computational Biology
---

## [Decision Letter · Decision Letter 2]

24 Oct 2023

Dear Dr. Poyatos,

We are pleased to inform you that your manuscript 'The limitations of phenotype prediction in metabolism' has been provisionally accepted for publication in PLOS Computational Biology. Three editors as well as the reviewers have evaluated the manuscript. While one of the reviewers has valid concerns regarding realistic media conditions and details of yeast physiology, we believe that the explicit statements in the manuscript regarding these issues - and that the focus of the manuscript is on the concept of using models to help understand and interpret gene association data - was a novel and creative use of computational models and will be useful for researchers.

Best regards,

William Cannon

Guest Editor

PLOS Computational Biology

Mark Alber

Section Editor

PLOS Computational Biology

Reviewer's Responses to Questions

**Comments to the Authors:**

Reviewer #1: An updated manuscript has been returned for review, and I can conclude from the point-to-point response that the revision mainly contributes with minor improvements to the original text of the manuscript.

Unfortunately, the Authors have not responded to the invitation of mine and Reviewer #3 to reflect more on the underlying biology in their study. On the contrary, the Authors took a defensive stance instead. Quoting Response to Reviewer #3:

"It is essential to note that our approach does not specifically address existing literature concerning specific biological findings, nor do we inquire about the direct applicability of our findings in a particular scenario. Instead, our emphasis lies on contributing valuable insights into the underlying biological mechanisms supporting the statistical predictions."

I believe it is not an appropriate response - and somewhat insulting - knowing that the manuscript is being considered in a publication with focus on life sciences.

However, this might be not the biggest issue I have with the revision. The Authors' response indicates that they are not in control of the knowledge and interpretation of existing work in (yeast) biology, something they try to approximate with computational models. I doubt that correct results can be built on incorrect foundations, I am sorry. In the Response, the Authors claim that they are, quote, "focusing on deriving a biological reasoning behind the precise value of R2 of a polygenic risk score (PGS)", which I have still failed to find in the Discussion of the paper. I am afraid that, while I had some issues with trusting the Authors' interpretation of the initial results, now I have even more of them because the hiccups in the Response point to sloppy biology in this work, which I find unacceptable.

Some highlights:

I have previously pointed out that the minimal media Authors simulate contains more glucose (20 mmol/gDW/h) than wild-type laboratory S. cerevisiae strains can consume in batch (around 16 mmol/gDW/h, Blank and Sauer 2004, PMID 15073318). First, the Authors mistreat the current consensus in the field on what a "minimal" medium is by blending in a custom definition of a "standard medium". By their description, the minimal media do not contain any carbon sources. Quote from Methods: "The minimal medium consists of the most fundamental components strictly necessary to support growth, albeit when supplemented with few other nutrients, it is defined by unlimited import and export of H2O, CO2, ammonia, phosphate, sulfate, sodium and potassium and is aerobic with an import rate of 2 mmol/gDW/h of O2."

More appropriate name for such a solution would be a buffer rather than media for a heterotrophic organism, since no growth would be possible in such "media". Their "standard" media description is what we call a minimal medium (single organic carbon source + inorganic salts). Authors should talk the same language as the audience. Authors argue that they took an established routine to derive media composition (Response: "This is an already validated protocol to mimic natural environments (for example, see reference Wang and Zhang 2009, lines 583-584).") but this should not prohibit from performing sanity checks, which are missing!

The Authors claim that they work only with respiratory phenotypes when, in their simulations, the oxygen uptake is severely restricted - they set the flux bound at 2 mmol/gDW/h. Quote from the Response: "Moreover, we indeed restrict the phenotype space to respiratory phenotypes by considering a realistic oxygen import rate. By doing so, we aim to focus on physiologically relevant metabolic behaviors that align with respiratory functions."

Have the Authors consulted at least one source on yeast growth?! First, S. cerevisiae can consume at least 4-fold more oxygen when growing under glucose limitation (van Hoek 1998, PMID 9797269), and second, oxygen consumption can be even higher when growing on other carbon sources, e.g. ethanol (Daran-Lapujade 2004, PMID 14630934, ethanol-limited chemostat cultures at ~60% maximal specific growth rate). Have the Authors tried to constrain the FBA model, which would have pointed that the yeast ferments most of the glucose into ethanol in their initial condition? This is definitely not (quote from Response) "plenty of oxygen and glucose in the unconstrained model"!

Final highlight of the Response comes from the discussion of generating rich media. Authors reply they controlled their media compositions not to get to too high specific growth rate. From Response: 'Our intention is not to avoid the use of "rich media" in general, but rather to address the issue of "unrealistically rich media" as in metabolic models one can generate arbitrarily large growth rates'. In the updated manuscript there is the following claim: "In addition, we discarded media that support biomass production rates <70% of those of the standard medium to avoid possible natural or model artifacts related to our implementation of quantitative mutations". So let's consider the following: take the condition with mmaximal uptake of glucose of 20 mmol/gDW/h, and oxygen of 2 mmol/gDW/h. Now, using the protocol, a number of neutral- or co-consumed C- or N-sources is added at uptake rates sampled from a distribution, and if the resulting medium can support >70% of the initial growth rate, it is discarded. I find this not only lacking any reasoning, but also - since Authors claim to care a lot about statistics - creating a huge bias in the dataset. To compare: typical lab strains of S. cerevisiae experimentally grow at specific growth rate mu=0.40/h in glucose-minimal media, and at around 0.47/h in very rich media formulations (YPD or SC). Very rich media give only a boost of some 20% growth! Thus, not analyzing formulations supporting mu < 0.28/h (70% of 0.40) discards a lot of realistic scenarios!

Side/technical note: I wonder why do Authors use a very old reconstruction of S. cerevisiae metabolism? iND750, when constrained with glucose and oxygen uptake bounds the Authors use, reports a predicted specific growth rate of 0.51/h (close to theoretical maximum, Metzl-Raz 2017 PMID 28857745). Yeast8, in contrast, predicts a growth rate closer to experimental observations (0.44/h).

Reviewer #2: The Authors have addressed my concerns.

**Have the authors made all data and (if applicable) computational code underlying the findings in their manuscript fully available?**

Reviewer #1: Yes

Reviewer #2: Yes

PLOS authors have the option to publish the peer review history of their article (what does this mean?). If published, this will include your full peer review and any attached files.

Reviewer #1: No

Reviewer #2: No

---

## [Editor Report · Acceptance letter]

4 Nov 2023

PCOMPBIOL-D-23-00225R2 

The limitations of phenotype prediction in metabolism

Dear Dr Poyatos,

I am pleased to inform you that your manuscript has been formally accepted for publication in PLOS Computational Biology. Your manuscript is now with our production department and you will be notified of the publication date in due course.

With kind regards,

Lilla Horvath
